# Feasibility and metabolic outcomes of a well-formulated ketogenic diet as an adjuvant therapeutic intervention for women with stage IV metastatic breast cancer: The Keto-CARE trial

**Alex Buga**[1], **David G. Harper**[2], **Teryn N. Sapper**[1], **Parker N. Hyde**[3], **Brandon Fell**[1], **Ryan Dickerson**[1], **Justen T. Stoner**[1], **Madison L. Kackley**[1], **Christopher D. Crabtree**[1], **Drew D. Decker**[1], **Bradley T. Robinson**[1], **Gerald Krystal**[4], **Katherine Binzel**[5], **Maryam B. Lustberg**[6], **Jeff S. Volek**[1] *

1 Department of Human Sciences, The Ohio State University, Columbus, Ohio, United States of America,
2 School of Kinesiology, University of the Fraser Valley, Abbotsford, British Columbia, Canada,
3 Department of Kinesiology, University of North Georgia, Dahlonega, Georgia, United States of America,
4 The Terry Fox Laboratory, BC Cancer Research Centre, Vancouver, British Columbia, Canada,
5 Department of Radiology, Wright Center of Innovation, The Ohio State University, Columbus, Ohio, United States of America, 6 Breast Cancer Center, Smilow Cancer Hospital, Yale University, New Haven, Connecticut, United States of America

* volek.1@osu.edu

**Data Availability Statement:** Data described in the manuscript will be made available immediately upon manuscript acceptance on the Ohio State

## Abstract

### Purpose

Ketogenic diets may positively influence cancer through pleiotropic mechanisms, but only a few small and short-term studies have addressed feasibility and efficacy in cancer patients. The primary goals of this study were to evaluate the feasibility and the sustained metabolic effects of a personalized well-formulated ketogenic diet (WFKD) designed to achieve consistent blood beta-hydroxybutyrate (βHB) >0.5 mM in women diagnosed with stage IV metastatic breast cancer (MBC) undergoing chemotherapy.

### Methods

Women ($n = 20$) were enrolled in a six month, two-phase, single-arm WFKD intervention (NCT03535701). Phase I was a highly-supervised, *ad libitum*, personalized WFKD, where women were provided with ketogenic-appropriate food daily for three months. Phase II transitioned women to a self-administered WFKD with ongoing coaching for an additional three months. Fasting capillary βHB and glucose were collected daily; weight, body composition, plasma insulin, and insulin resistance were collected at baseline, three and six months.

### Results

Capillary βHB indicated women achieved nutritional ketosis (Phase I mean: 0.8 mM ($n = 15$); Phase II mean: 0.7 mM ($n = 9$)). Body weight decreased 10% after three months,

Dryad database at: https://doi.org/10.5061/dryad.
kh18932d4.

**Funding:** Grant Recipient: JV Grant Award #: AWD-
102893 Grant Organization: Lotte and John Hecht
Memorial Foundation Grant Website: https://www.
hecht.org/ The funders had no role in study design,
data collection and analysis, decision to publish, or
preparation of the manuscript.

**Competing interests:** I have read the journal's
policy and the authors of this manuscript have the
following competing interests: JSV receives
royalties from book sales; is a founder and has
equity in Virta Health; and is a science advisor for
Simply Good Foods and Cook Keto. MBL has
received consulting fees from AstraZeneca,
Biotheranostics, Novartis, Pfizer and PledPharma.
The remaining authors have no relevant financial or
non-financial interests to disclose. This does not
alter our adherence to PLOS ONE policies on
sharing data and materials.

primarily from body fat. Fasting plasma glucose, plasma insulin, and insulin resistance also decreased significantly after three months ($p < 0.01$), an effect that persisted at six months.

## Conclusions

Women diagnosed with MBC undergoing chemotherapy can safely achieve and maintain nutritional ketosis, while improving body composition and insulin resistance, out to six months.

## Introduction

In 2020, breast cancer became the most diagnosed cancer worldwide–with an estimated 2.3 million new cases–surpassing lung cancer [1]. Early detection and treatment have improved breast cancer survival rates to almost 90% [2], however, approximately only one in three women with stage IV metastatic breast cancer (MBC) survive past five years [3].

Previous dietary interventions, exploring MBC outcomes, focused on low-fat/high-carbo-hydrate regimens that produced disappointing results [4–6] indicating limited efficacy. Obser-vational studies examining the role of fat in breast cancer have also demonstrated mixed results [7, 8]. The ambient dietary carbohydrate intake is likely to be an important consider-ation in how dietary fat is metabolized, which in turn might explain inconsistent associations between fat consumption and cancer risk. We previously demonstrated that in the context of a carbohydrate-restricted well-formulated ketogenic diet (WFKD), increased consumption of saturated fat does not lead to increased levels of circulating saturated fat; in fact, blood satu-rated fat usually decreases [9–12]. Beyond positively influencing saturated fat metabolism, a WFKD is associated with broad-spectrum health benefits, many of which might have favorable impact on breast cancer [13] treatment and outcomes.

The daily carbohydrate restriction (20 – 50g/day) inherent to a WFKD leads to a reduction in plasma insulin and glucose levels, thereby increasing reliance on fatty acid oxidation over glucose, which may favorably influence local tumor and systemic metabolism. Ketogenic diets may also improve cancer outcomes through additional pleiotropic mechanisms such as alter-ing tumor metabolism (i.e., downregulate aerobic glycolysis; Warburg effect), improving mito-chondrial hormesis, restoring normal oxidative phosphorylation, influencing epigenetic modulation, reducing 'cancer-related upregulation of phosphoinositol-3 kinase (PI3K) and mammalian target of rapamycin (mTOR) signaling, and lessening hyperglycemia induced by cancer medication [14–18].

While there are many hypothetical reasons a metabolic state of nutritional ketosis may ben-efit individuals with cancer, there are only a few published studies demonstrating feasibility and metabolic outcomes in humans. These include case studies [19, 20] and preclinical models [21–23]; only a few involved longitudinal studies in breast cancer populations [24–27]. These small trials reported that ketogenic diets are safe and effective at improving fasting glucose, insulin [26], body composition [24, 25], and quality of life [27]. There are no previous studies on MBC patients lasting longer than three months that monitored plasma glucose and ketones daily to track adherence and guide sustainability.

The KETOgenic diet in Chemotherapy to Affect REcurrence of breast cancer (Keto-CARE) trial is among the few exploratory safety trials designed to investigate the feasibility and meta-bolic effects of a WFKD in women with stage IV MBC undergoing chemotherapy. A major focus was to design the WFKD intervention to be enjoyable and sustainable by tailoring it for

each participant with the goal of achieving and maintaining nutritional ketosis throughout six-months. Feasibility was evaluated by monitoring safety and attrition. Dietary adherence was defined by daily capillary beta hydroxybutyrate (βHB) concentrations maintained within a confined nutritional ketosis range (0.5–4.0 mM βHB). Phase I (highly supervised; months 0–3) provided all the food and coaching for the WFKD; phase II (self-administered; months 3–6) provided coaching only. Blood parameters such as fasting plasma glucose and insulin, and body composition were assessed at baseline and after each phase.

## Methods

### Study participants

Participants were recruited through the Stefanie Spielman Comprehensive Breast Center at The Ohio State University from October 2017 to September 2019. Data collection and last the last follow-up ended in March 2020. The goal was to enroll 20 women with diagnosed MBC, currently undergoing standard-of-care anticancer therapies, into a WFKD intervention. A summary of patients' specific disease subtypes (i.e., hormone phenotype), anatomical sites of metastases, and treatment regimens are summarized in Table 1. Detailed drug descriptions are summarized in the supplement (S1 Table). To maximize recruitment and sample size require-ments, the trial did not limit participants to specific histological subtypes and/or treatment regimen. A list of inclusion and exclusion criteria are presented in Table 2. The study was

**Table 1. Participant descriptives.**

| Treatment | Subtype | Metastases | Age | BMI | Total Body Fat (%) |
|---|---|---|---|---|---|
| | ER PR HER2 | | | | |
| Capecitabine | - - - | LU, BO, BR | 52 | 41 | 53.1 |
| Capecitabine | + + - | LI, BO | 48 | 24 | 36.4 |
| Capecitabine | - - - | BO, LI | 67 | 22 | 41.4 |
| Capecitabine | - - - | LI | 57 | 23 | 45.8 |
| Capecitabine | - - - | BO, LU, BR | 60 | 29 | 49.9 |
| Capecitabine | - - - | LU | 55 | 30 | 41.9 |
| Capecitabine | + + - | BO | 72 | 44 | 53.9 |
| Capecitabine | - - - | LU, LY, BO | 60 | 22 | 42.3 |
| Capecitabine/Paclitaxel | + + - | BO, PE | 87 | 24 | 37.4 |
| Docetaxel/Peruzumab/Trastuzumab | + + + | LI, BO | 57 | 33 | 45.7 |
| Doxorubicin | + + - | LI, BO | 51 | 26 | 37.4 |
| Exemestane | + + - | BO, LI | 60 | 26 | 43.8 |
| Paclitaxel | + + - | LI, BO | 43 | 26 | 34.5 |
| Paclitaxel | +—+ | LI | 53 | 35 | 52.1 |
| Paclitaxel | +—- | LI, BO, LU | 61 | 25 | 41.5 |
| Paclitaxel | - - - | LU, LY, LI | 59 | 31 | 43.5 |
| Palbociclib | + + - | BO | 68 | 31 | 49.5 |
| Palbociclib | + + - | BO | 51 | 35 | 44.0 |
| Palbociclib/Letrozole/Denosumab | + + - | BO | 39 | 38 | 51.7 |
| Palbociclib/Letrozole/Denosumab/Goserelin | + + - | BO | 35 | 46 | 54.5 |
| **Mean ± SD** | | | 57 ± 12 | 31 ± 7 | 45.0 ± 6.2 |

ER, estrogen receptor; PR, progesterone receptor; HER2, human epidermal growth factor receptor-2

BO, bone; BR, brain; LI, liver; LU, lung; LY, lymph; PE, peritoneum.

**Table 2. Inclusion/exclusion criteria.**

| Inclusion Criteria | Exclusion Criteria |
|---|---|
| • Age $\geq$ 18 years<br>• Body mass index (BMI) $\geq$ 22 kg/m$^2$<br>• Confirmed diagnosis of metastatic or stage IV breast cancer<br>• FDG-PET avid tumors<br>• Eastern Cooperative Oncology Group (ECOG) Performance status of 0–1 (0 = participant has normal activity, 1 = participant has some symptoms but is nearly full ambulatory)<br>• Able and willing to follow prescribed diet intervention.<br>• Life expectancy > 6 months | • Greater than 3 rounds of prior chemotherapy for MBC (prior to adjuvant chemotherapy permitted as long as > 12 months)<br>• BMI <22 kg/m$^2$<br>• Weight change > 5% within 3 months of enrollment<br>• Type 1 diabetes<br>• Current use of insulin or sulfonylureas for glycemic control or history of ketoacidosis<br>• History of diabetes with retinopathy requiring treatment<br>• Intestinal obstruction<br>• Abnormal liver or kidney function<br>• Congestive heart failure<br>• Pregnant or nursing<br>• Uncontrolled medical conditions that would limit compliance with study requirements.<br>• Unable to provide informed consent |

registered on ClinicalTrials.gov (NCT03535701) prior to commencing data collection. Approvals for the clinical protocols and clinical oversight were provided by The Ohio State University Institutional Review Board (IRB; #2017C0020) and the Clinical Scientific Review Committee (CSRC; OSU-16289) at the OSU Comprehensive Cancer Center. Written informed consent was obtained from all individual participants included in the study, prior to commencing data collection, and in accordance with the *Declaration of Helsinki 2013*. A detailed CONSORT diagram depicts the study enrolment (Fig 1).

## Experimental design

Keto-CARE was originally designed as a two-arm, biphasic trial aimed at establishing the feasibility and clinical use of a WFKD compared to the standardized control diet for women with MBC. To address our primary aim of adherence and feasibility, we used daily fasting capillary βHB trends over six months as the main outcome. The secondary exploratory outcomes included the effects of WFKD on glucose, insulin, body weight, and body composition.

Participants had the choice to self-select into either group following standard dietary recommendations as reported by the World Cancer Research Fund/American Institute for Cancer Research (WCR/AICR) [28]. Two major events prevented the enrollment into the control group. First, all the eligible participants self-selected into the WFKD dietary arm. Second, by the time we focused on recruitment to the control group, the COVID-19 pandemic led to a sustained cessation of clinical trials. Therefore, the study was redesigned into a single-arm intervention, similar to other small sample size oncology studies [26, 29–32] (Fig 2).

During the first, three month phase of the study (Phase I), team registered dietitians provided "highly-supervised" dietetic support to participants as daily WFKD coaching and nutrition guidance, and similar to previous continuous-care models [33]. To overcome possible WFKD challenges (i.e., ketosis maintenance) and attrition described by a similar trial [34], the grant award from the Hecht Foundation provided the funds necessary to rent out a metabolic kitchen space, purchase all the raw-food items from local grocery stores, and cook/assemble daily breakfasts, lunches, dinners, and snacks for participants throughout Phase I. During the

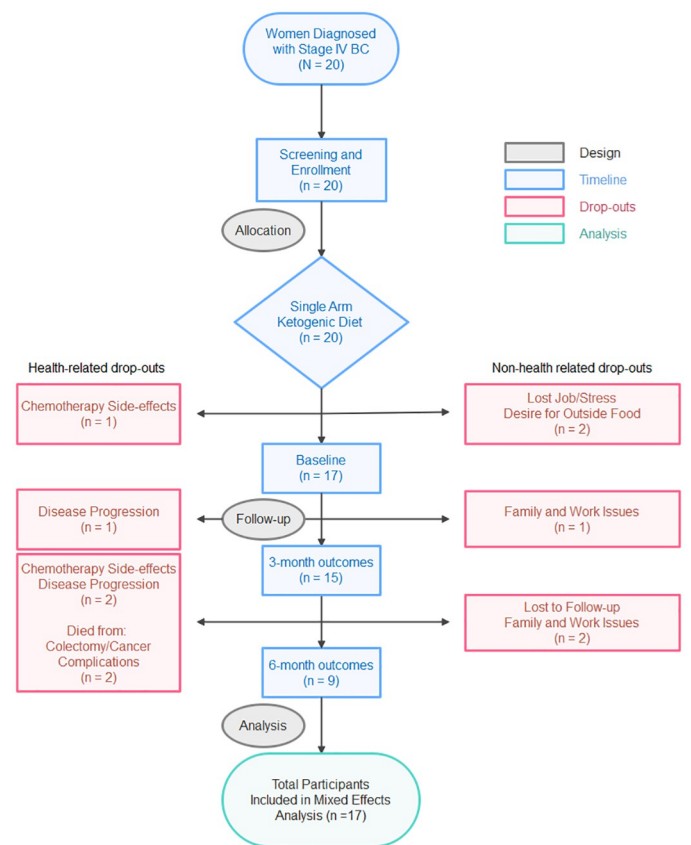

**Fig 1. CONSORT diagram.**

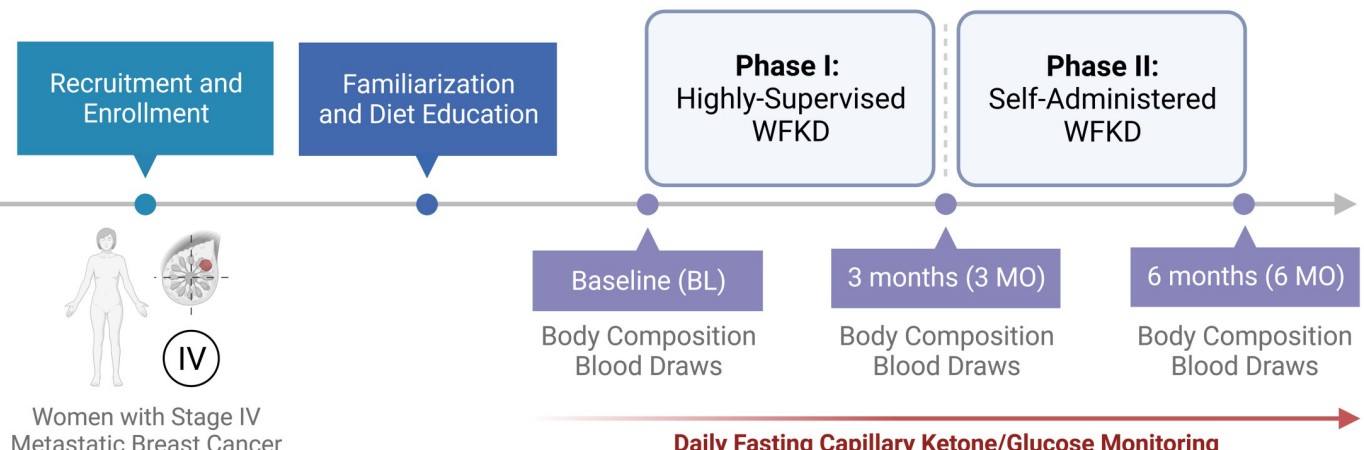

**Fig 2. Experimental design.** Women diagnosed with stage IV metastatic breast cancer ($n$ = 20) were recruited and assigned to participate in a single-arm, well-formulated ketogenic diet (WFKD) intervention. The highly supervised WFKD portion (Phase I) provided three months of food and counseling from a specialized ketogenic-trained registered dietitian. The self-administered WFKD (Phase II) consisted of a 3-month free-living period where the feasibility of consuming a WFKD without food provision was evaluated. Figure was created with BioRender.com.

second, three month phase of the study (Phase II), the study team remained in daily contact with participants to monitor plasma ketones and glucose, and participants were responsible for maintaining a "self-administered" WFKD while receiving continued coaching and guidance as requested.

All participants received their standard-of-care chemotherapy treatment, prescribed by their oncologist, most often paclitaxel or capecitabine. The tumor subtypes of the group included estrogen (ER) positive/negative, progesterone (PR) positive/negative, hormone estrogen receptor (HER) positive/negative, and combinations of those subtypes.

### Dietetic intervention

**Phase I: Highly supervised WFKD.** Prior to initiating the diet intervention, participants met with registered dietitians to receive education on the principles of a WFKD and ketone metabolism. The individualized WFKD consisted of 20–50 g/day of carbohydrate, 1.2–1.5 g of protein/kg of reference weight [35], and dietary fat consumed to satiety. All meals were prepared fresh in a state-of-the-art metabolic kitchen (Instructional Research Kitchen, Ohio Union, Columbus, OH); frozen meals were provided by a private supplier (PangeaKeto, North Canton, OH). Additional groceries were purchased by the study team based on individual preferences and requirements. All meals and groceries were provided *ad libitum* and customized to individual participant preference. The primary benchmark of Phase I was to achieve blood ketones ≥0.5 mM βHB, as determined by fasting capillary finger sticks performed each morning and alike to prior methods [36]. In addition to the initial education session, the study team was in daily communication with participants via phone, emails, and/or text messages, to monitor adherence and support participants in maintaining nutritional ketosis.

A wide variety of foods were incorporated into the nutritional program including non-starchy vegetables (e.g., leafy greens, broccoli, cauliflower, zucchini), low-glycemic fruits (e.g., berries, olives, tomatoes, lemons/limes), animal-based protein (e.g., eggs, beef, poultry, pork, seafood), nuts, seeds, oils (olive, avocado, coconut), high-fat dairy (cheese, butter, cream). Saturated and monounsaturated fats were the primary fat sources, while polyunsaturated fats were consumed in low to moderate amounts. Participants were encouraged to increase their sodium intake by 1–2 g/day by liberally salting foods, drinking bouillon or broth daily, or making an electrolyte beverage to avoid hyponatremia while in nutritional ketosis [37]. To increase tolerability and sustainability, we did not require tracking of daily macronutrients and calories. Participants were instructed to eat to satiety and were allowed to lose, gain, or maintain weight as desired and as advised by their attending medical oncologist. In addition to the unlimited amounts of fresh and frozen study foods, participants were free to eat ketogenic-appropriate foods outside of what the study provided (e.g., at restaurants, work, or family gatherings). A sample menu template is depicted in Fig 3 and a summarized list of food items is provided in the supplement (S2 Table).

At the beginning of Phase I, participants relied heavily on the fresh and frozen meals provided by the study to achieve nutritional ketosis. As participants demonstrated that they could maintain capillary ketone concentrations of ≥0.5 mM βHB, they transitioned to a hybrid approach where the study team provided them with weekly groceries and meal/recipe guidance to prepare WFKD meals at home. The transition from ready-to-eat meals to weekly grocery procurement was intended to prepare women for Phase II of the study.

**Phase II: Self-administered WFKD.** For the second 3-month experimental phase participants 'self-administered' the WFKD, allowing us to investigate the feasibility of maintaining the dietary approach without the intense supervision and food provision as provided in Phase I. Participants were encouraged to follow the same WFKD principles as Phase I. Food was no

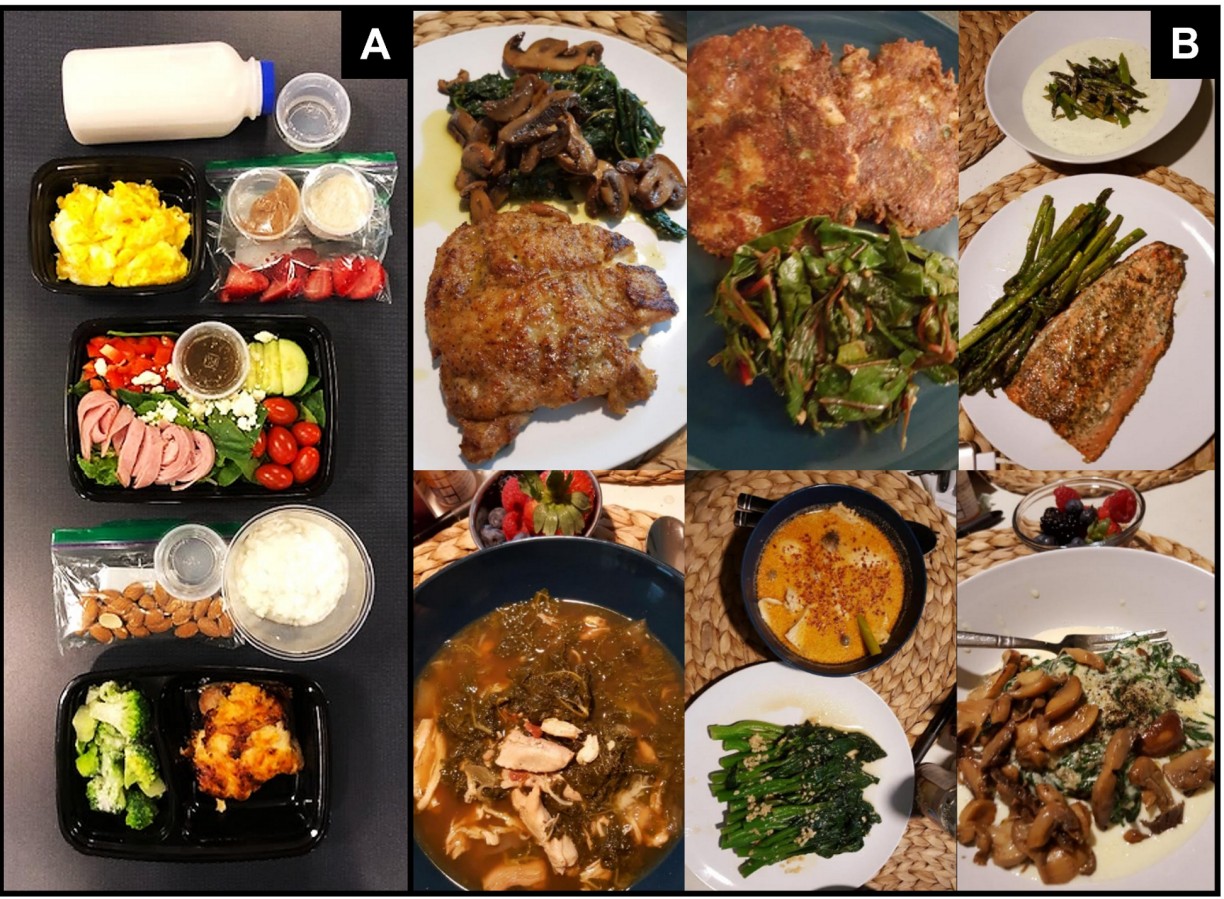

**Fig 3. Menu template.** Panel A: Sample breakfast, lunch, dinner, and snacks (~2000 kcal) prepared in the metabolic kitchen during the highly-supervised study phase (Phase I). Panel B: Sample lunches and dinners prepared by participants, at home, during the self-administered diet feasibility evaluation phase (Phase II). Featured meals comprise salmon with cream of asparagus soup, chicken with sautéed mushrooms, zucchini cakes, beef soup, chicken and squash soup, and mushroom risotto.

longer provided, but the research team was available for daily communication with the participants to help facilitate the maintenance of nutritional ketosis. Daily fasting capillary βHB and glucose continued to be reported during this phase.

## Metabolic measurements, blood collection and analysis

Fasting βHB and glucose were measured daily–in the morning and before breakfast–in capillary blood using enzymatic reagent strips fitted for a portable Precision Xtra$^{©}$ glucometer/ketometer (Abbott, Inc., Columbus, OH). Participants reported their values by submitting pictures of the meter readings to study staff to minimize self-reporting bias. Fasting blood samples were collected by a trained phlebotomist via venipuncture in the antecubital fossa (Eppendorf 21G Butterfly, Hamburg, Germany) or port access. All participants reported to the testing facility before 9:00am for their assessments (Martha Morehouse, Columbus, OH). Arrival conditions stipulated that subjects consume no caffeine for >12h, no food for >8h, and sleep 6-8h the night before testing. Blood was collected into a single 10mL spray coated $K_2$EDTA vacutainer tube (BD Vacutainer, Franklin Lakes, NJ). Blood collection tubes were then centrifuged at 1200 g for 10 minutes at 4˚C. Plasma was aliquoted and snap-frozen in liquid nitrogen.

Samples were stored at -80˚C and thawed immediately prior to analysis. Serum insulin was determined via enzyme-linked immunosorbent assay (ELISA) with an average within-samples CV of 2.1% (Quantikine, Catalog No. DINS00, R&D Systems Minneapolis, MN). Insulin resistance (HOMA-IR) was determined from capillary glucose and plasma insulin [38].

### Body composition

Body weight and composition were analyzed at baseline and end of each phase. Weight was measured at each visit on an electronic stadiometer (SECA 703 Digital, Hamburg, Germany) calibrated to the nearest ±0.01 kg while participants wore light clothing and no shoes. Body composition was assessed by a licensed technician using dual-energy X-ray absorptiometry (Lunar iDXA; GE, Madison, WI, USA). Regional fat mass and lean body mass were quantified from a whole-body scan at each visit (CoreScan™ enCORE software version 14.10).

### Statistical analysis

Analyses and graphs were performed using GraphPad Prism (ver. 9.1.0, GraphPad Software, San Diego, California USA). Two-tail $\alpha$ significance was set at $p < 0.05$. All the variables of interest were screened for normality and homogeneity of variance using the Shapiro-Wilk test and Mauchly's sphericity test, respectively. Violations of sphericity were treated with the Greenhouse-Geisser adjustment. For Phase I endpoints, we analyzed the main effects of time using paired samples t-tests (BL vs. 3 MO). When including Phase II endpoints, we used a 1 (condition) x 3 (time) repeated-measures mixed-effects analysis of variance (RM ANOVA) with Bonferroni post-hoc corrections to analyze the differences between BL and 3 MO; BL and 6 MO; and 3 MO and 6 MO. We additionally compared weekly fasting glucose/βHB mean concentrations and variability between phases using an unpaired t-test (Phase I vs. Phase II).

We planned to recruit 20 participants to provide a reasonable representation of variability in diet adherence and metabolic outcomes to inform future studies. Additionally, due to the novel nature of this protocol we also anticipated substantial attrition and baseline variance. Due to a lack of comparison arm and the fact that non-compliance (as guided by capillary βHB) was a primary outcome, we did not perform an intent-to-treat analysis. To preserve power, we instead analyzed participants based on their enrollment in the study as follows: those who completed the study up to and including the three-month outcome were treated as "Phase I only" participants; those who completed the six-month outcome were treated as "Phase I+II" completers. Detailed information regarding the study design and de-identified data are publicly available in the supplement (S1 and S2 Files and the online dataset [39].

## Results

### Enrollment and participant characteristics

Out of the 20 enrolled participants, 5 dropped out during Phase I, and 6 prior to completing Phase II. Thus, 15 participants completed Phase I and 9 participants completed both Phase I and II. Baseline characteristics, disease subtype, study duration, chemotherapy regimen, and brief rationale for withdrawal are presented in Fig 4. In no instance was the primary reason for subject withdrawal attributed to the WFKD. There were also no adverse events associated with the adoption or maintenance of the WFKD.

### Phase I: Highly supervised WFKD

**Capillary ketones and glucose.** Results for Phase I are summarized in Table 3, with individual results shown for blood markers (Fig 5). Participants reported fasting capillary βHB and

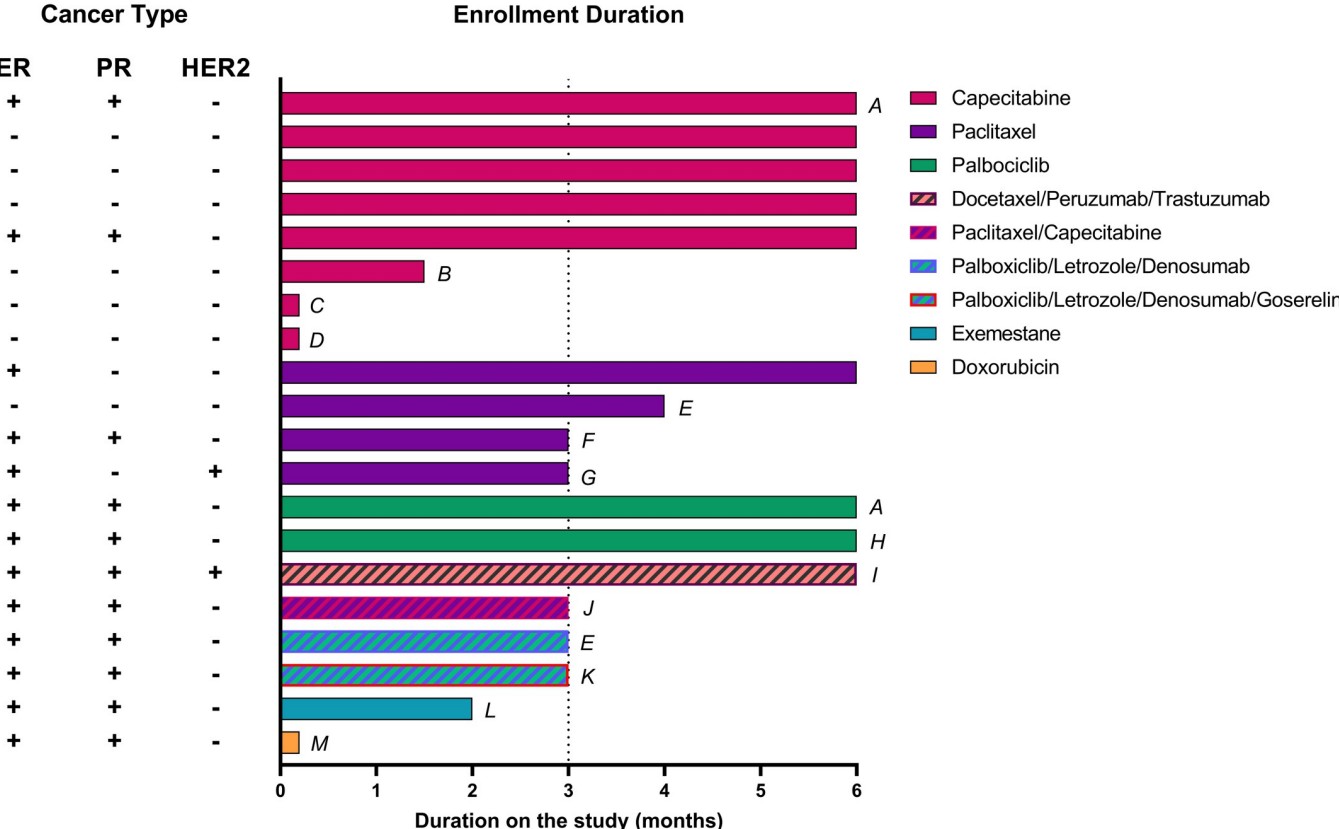

**Fig 4. Enrollment Gantt chart.** A: completed six-months of WFKD but failed to report back to study center for post-testing. B: lost job/stress of treatment regimen. C: early side effects of chemotherapy. D: progressive disease before starting trial. E: progressive disease. F: family problems/challenges. G: did not maintain contact with study team. H: completed six-months of WFKD but was restricted from research center for post-testing due to COVID I: completed six-months of WFKD but passed away from cancer-related complications before post testing. J: passed away after surgery. K: chose not to maintain ketosis. L: family and work issues. M: preference for non-ketogenic desserts as comfort food.

**Table 3. Results and main effects during the highly-supervised WFKD intervention.**

| Variable | n | Baseline | | Phase I | | Δ | % Δ | Effect Size (Cohen's *d*) | Paired t-test (*p*-value) |
|---|---|---|---|---|---|---|---|---|---|
| | | Mean | SEM | Mean | SEM | | | | |
| Ketones (βHB; mM) | 15 | 0.18 | 0.02 | 0.81 | 0.11 | 0.63 | 450% | 1.99 | **0.001** |
| Glucose (mg/dL) | 15 | 119 | 5.3 | 105 | 2.6 | -14 | -12% | 0.86 | **0.009** |
| Insulin (µU/mL) | 14 | 24.5 | 3.5 | 17.0 | 1.8 | -7.5 | -31% | 0.72 | **0.008** |
| HOMA-IR | 14 | 7.4 | 1.0 | 4.5 | 0.5 | -2.9 | -39% | 0.94 | **0.002** |
| Body Weight (kg) | 15 | 86.4 | 4.9 | 77.5 | 4.5 | -8.9 | -10% | 0.49 | **0.001** |
| Body Fat Mass (kg) | 15 | 39.4 | 3.7 | 32.7 | 3.4 | -6.7 | -17% | 0.49 | **0.001** |
| Lean Body Mass (kg) | 15 | 46.9 | 1.5 | 44.8 | 1.4 | -2.1 | -4% | 0.38 | **0.006** |
| Body Fat Percentage (%) | 15 | 44.5 | 1.7 | 40.9 | 1.8 | -3.6 | | 0.55 | **0.001** |

n = 1 blood sample was excluded for Insulin/HOMA-IR measurement due to hemolysis.

Effect size thresholds: 0.2 –weak effect; 0.5 –moderate effect; 0.8 –large effect.

Δ = change from BL. Bold face denotes significant effects.

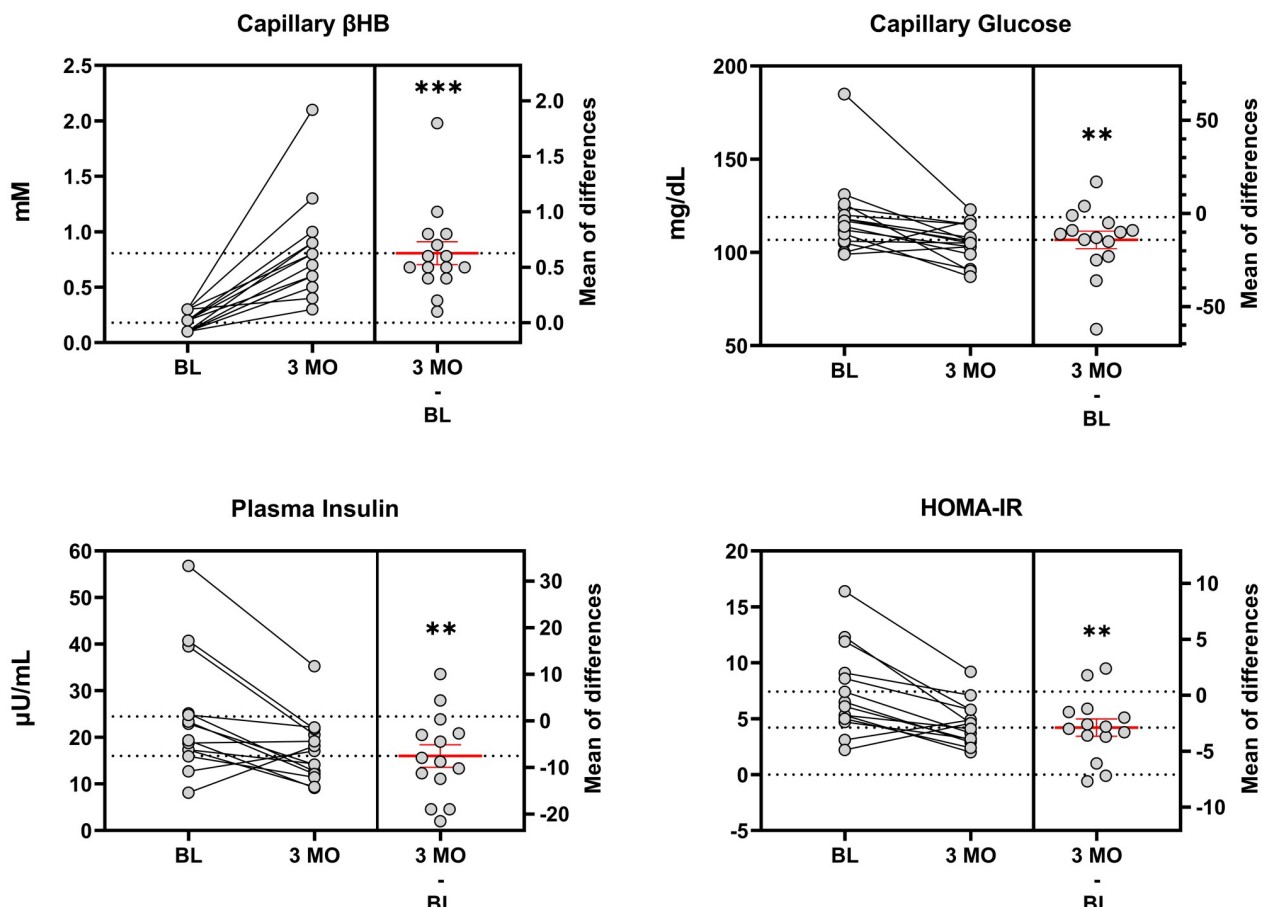

**Fig 5. Blood parameters.** Baseline (BL) and three month (3 MO) data points for ketones and glucose are reported as the test visit capillary blood measures collected during the clinically controlled WFKD phase. Insulin was measured in fasting plasma and quantified using ELISA. Data presented as mean ± SEM. Time effects:**,*** < 0.01/0.001 from BL. HOMA-IR, homeostatic model of insulin resistance. HOMA-IR = [glucose (mg/dL) x insulin (μU/mL)] / 405.

glucose readings to the staff 92% of the days during Phase I. All participants ($n$ = 15) attained nutritional ketosis after the first week on WFKD and maintained βHB ≥0.5 mM 87% of the time over the three months. Mean fasting βHB increased over three months, and into the predicted range of nutritional ketosis (Δ: 0.6 ± 0.1 mM, $p < 0.001$).

Fasting capillary glucose decreased 12% from baseline (Δ: -14 ± 5 mg/dL, $p = 0.009$). One participant with hyperglycemia demonstrated a 62 mg/dL decrease in glucose (-34%); the remaining participants demonstrated a more modest but still significant decrease in glucose (mean -11 mg/dL, 95% CI: -18 to -3 mg/dL; $p < 0.01$).

**Plasma insulin.** Fasting plasma insulin decreased 30% from baseline (Δ: -7 ± 2 μU/mL; $p = 0.007$), which corresponded to a 38% decrease in HOMA-IR (Δ: 2.7 ± 0.7; $p = 0.002$). Thirteen participants (87%) experienced an improvement in insulin sensitivity.

**Body weight and composition.** All participants lost body weight and body fat mass during Phase I (Fig 6). On average, weight decreased by 10% from baseline (Δ: -8.5 ± 1.4 kg; $p > 0.001$). Mean body composition changes were mainly driven by a decrease in body fat percentage (Δ: -3.7 ± 0.5%; $p < 0.001$). 'The overall ratio of fat mass to lean body mass lost was 3.5:1 kg:kg, with 78% of the weight loss derived from body fat mass (Δ: -6.6 ± 1.0 kg; $p > 0.001$) and 22% from lean body mass (Δ: -1.8 ± 0.7 kg; $p = 0.02$).

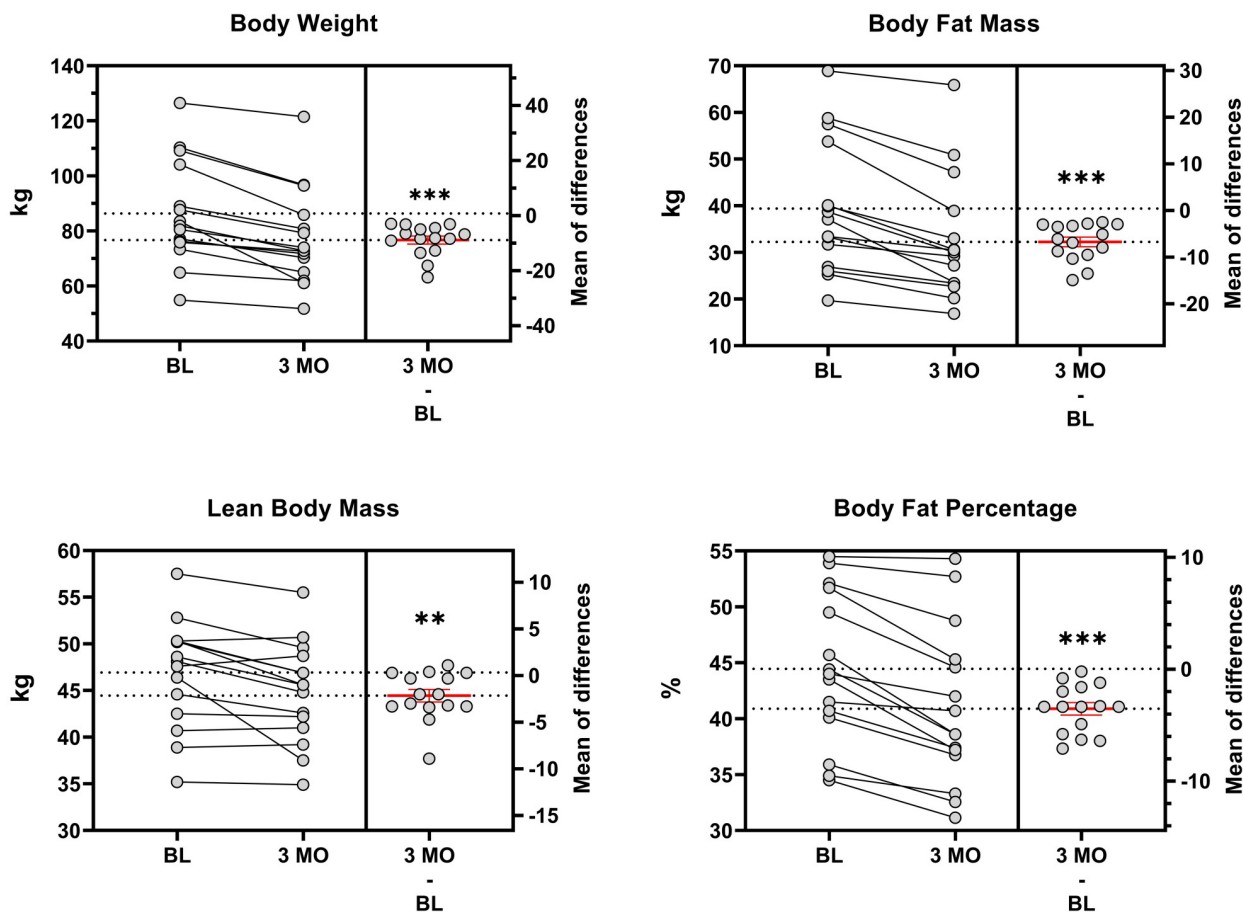

**Fig 6. Body composition.** Data presented as mean ± SEM. Time effect:.**,*** < 0.01/0.001 from BL.

## Phase II: Self-administered WFKD

**Capillary ketones and glucose.** Phase II outcomes are summarized in Tables 4 and 5. Participants reported a weekly fasting βHB ≥ 0.5 mM for 93% of the daily capillary measurements. When examining within-group differences both phases demonstrated significantly elevated ketones from baseline ($p < 0.001$). There were no differences between blood βHB in the women who only completed Phase I and those who completed both Phase I+II (0.7 ± 0.1 vs. 0.6 ± 0.1; $p = 0.12$).

Weekly fasting capillary glucose values did not decrease from the beginning of phase I and throughout Phase II. Between-group comparisons revealed that participants who completed Phase I had a 7 mg/dL higher fasting blood glucose (110 ± 1 vs. 103 ± 0.4 mg/dL; $p < 0.001$) and more than twice the variability (7% vs. 16%; $p < 0.001$) compared to Phase I+II completers (Fig 7).

**Blood parameters and body composition.** Of the nine women who finished Phase II, four did not complete all the requirements for the blood parameters and body composition testing. Therefore, only five women were included in the Phase I+II analysis. Capillary βHB remained elevated after Phase I and thereafter and was significantly greater throughout both phases compared to baseline. Capillary glucose and insulin sensitivity improved after Phase I and remained significantly lower thereafter compared to baseline. Body weight, fat mass, and

**Table 4. Monthly capillary ketone and glucose variability.**

| Variable | Completion Date | Completed Phase I Only | | | | Completed Phase I+II | | | | Totals |
|---|---|---|---|---|---|---|---|---|---|---|
| | | *n* | Mean | 95% CI | | *n* | Mean | 95% CI | | *n* |
| | | | | Lower | Upper | | | Lower | Upper | |
| Ketones (βHB; mM) | 1 Month | 8 | 0.90 | 0.40 | 1.40 | 9 | 0.78 | 0.42 | 1.14 | 17 |
| | 2 Months | 6 | 0.61 | 0.29 | 0.93 | 9 | 0.73 | 0.40 | 1.05 | 15 |
| | 3 Months | 5 | 0.43 | 0.06 | 0.79 | 9 | 0.75 | 0.49 | 1.01 | 14 |
| | 4 Months | | | | | 9 | 0.65 | 0.47 | 0.83 | 9 |
| | 5 Months | | | | | 9 | 0.61 | 0.45 | 0.77 | 9 |
| | 6 Months | | | | | 9 | 0.60 | 0.41 | 0.79 | 9 |
| Glucose (mg/dL) | 1 Month | 8 | 111 | 97 | 125 | 9 | 104 | 97 | 110 | 17 |
| | 2 Months | 6 | 111 | 93 | 130 | 9 | 102 | 97 | 107 | 15 |
| | 3 Months | 5 | 107 | 87 | 126 | 9 | 103 | 98 | 108 | 14 |
| | 4 Months | | | | | 9 | 104 | 97 | 111 | 9 |
| | 5 Months | | | | | 9 | 101 | 94 | 107 | 9 |
| | 6 Months | | | | | 9 | 100 | 93 | 107 | 9 |

lean body mass decreased non-significantly after Phase I but reached significance after Phase II. Body fat percentage decreased from baseline non-significantly.

## Discussion

We demonstrated that a three-month highly supervised WFKD, followed by a three-month self-administered WFKD, was a well-tolerated and sustainable dietary approach for most women with MBC undergoing chemotherapy. Nutritional ketosis was achieved in all the women who completed the first three months, facilitated by using prepared/procured food and frequent nutrition coaching guided by daily monitoring of fasting βHB and glucose. After three months participants received less intense coaching, but maintained nutritional ketosis, improved body weight, body composition, glucose, and insulin sensitivity. To our knowledge, this is the first study to demonstrate that nutritional ketosis can be safely sustained with proper nutritional guidance and monitoring in women with MBC undergoing chemotherapy over six-months.

**Table 5. Results and main effects during the highly supervised and self-administered WFKD intervention.**

| Variable | *n* | Baseline | | Phase I | | Phase II | | Post-hoc comparisons (*p*-value) | | |
|---|---|---|---|---|---|---|---|---|---|---|
| | | Mean | SEM | Mean | SEM | Mean | SEM | BL vs. Phase I | BL vs. Phase II | Phase I vs. Phase II |
| Ketones (βHB; mM) | 5 | 0.20 | 0.0 | 0.74 | 0.2 | 0.72 | 0.1 | **0.005** | **0.006** | 0.99 |
| Glucose (mg/dL) | 5 | 119 | 4.5 | 102 | 3.2 | 103 | 2.6 | **0.035** | **0.040** | 0.99 |
| Insulin (µU/mL) | 5 | 26.6 | 5.7 | 17.6 | 1.6 | 14.8 | 2.8 | 0.087 | **0.026** | 0.99 |
| HOMA-IR | 5 | 7.6 | 1.5 | 4.5 | 0.5 | 3.8 | 0.7 | **0.017** | **0.005** | 0.99 |
| Body Weight (kg) | 5 | 76.7 | 3.3 | 67.6 | 2.5 | 65.9 | 1.6 | 0.059 | **0.026** | 0.99 |
| Body Fat Mass (kg) | 5 | 31.0 | 2.1 | 25.2 | 1.3 | 23.8 | 1.6 | 0.090 | **0.034** | 0.99 |
| Lean Body Mass (kg) | 5 | 45.7 | 2.0 | 42.4 | 1.8 | 42.1 | 1.6 | 0.061 | **0.041** | 0.99 |
| Body Fat Percentage (%) | 5 | 40.3 | 1.5 | 37.4 | 1.2 | 36.1 | 2.1 | 0.246 | 0.070 | 0.99 |

All p-values were generated from a 1x3 RM ANOVA using the Bonferroni correction for multiple comparisons.
Bold face denotes p < 0.05.

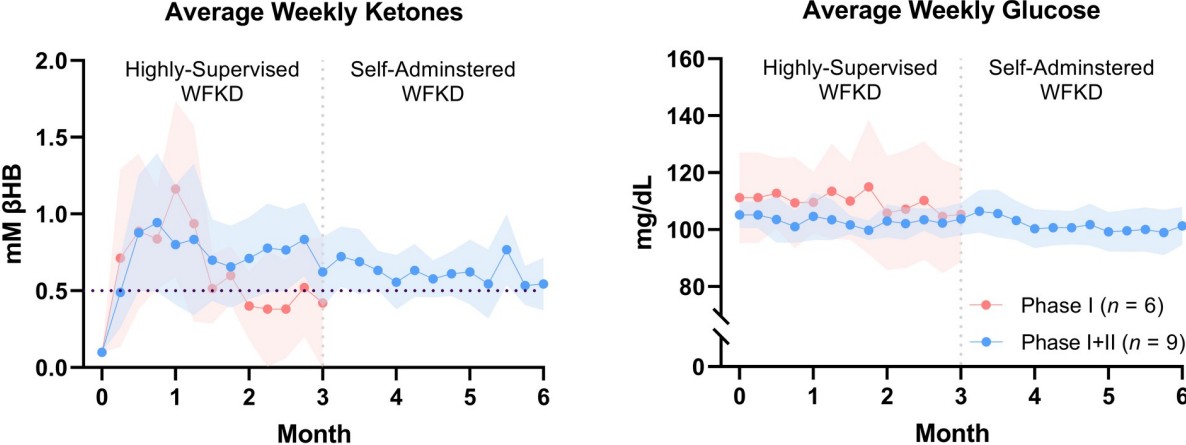

**Fig 7. Weekly fasting capillary βHB and glucose.** Each data point represents a 7-day fasting capillary blood collection average. The clouds surrounding the moving average denote the 95% confidence interval.

Consistent with our results, prior studies that implemented a ketogenic diet in breast cancer patients reported maintenance of nutritional ketosis as guided by capillary ketones [24–26]. The attrition rate after Phase I (47%) and Phase II (75%) is higher than our previous studies in non-cancer populations [11, 34, 40, 41], but matched expectations from a previous WFKD feasibility trial in cancer populations that reported ~70% attrition over three months [31]. Our feasibility results are positively corroborated by non-breast cancer studies that established carbohydrate-restriction feasibility in glioblastoma, melanoma, endometrial, and several other advanced forms of cancer [32, 42–44], thereby strengthening our exploratory approach to breast cancer feasibility and adherence, specifically in women with stage IV MBC. Importantly, the reasons for withdrawal from the study were not due to the switch to a WFKD, but instead were attributed to medical and non-medically related complications described in greater detail on Fig 2. Irrespective of study duration women diagnosed with MBC demonstrated a strong interest in understanding the principles of WFKD and adhering to the diet over several months as guided by daily capillary βHB.

After nutritional ketosis was attained (βHB ≥0.5 mM), participants remained in this metabolic state 90% of the time, demonstrating excellent adherence to the ketogenic diet. This is especially noteworthy as chemotherapy sessions and concomitant administration of steroids are associated with hyperglycemia and an overall anti-ketogenic effect [14, 26, 45–48]. The frequency of capillary βHB reporting agrees with prior maintenance of nutritional ketosis maintenance in short-term cancer trials [26, 45]. Daily capillary BHB objectively assessed dietary adherence and that data was then used by the team to adjust meal composition (e.g., adjust carbohydrate and/or protein intake) to ensure that βHB concentrations remained in nutritional ketosis range [49]. Dietary adjustments were of particular importance in this study because they reveal the feasibility of clinical guidance and help minimize confounding diet effects. Our daily capillary βHB monitoring model and continuous dietitian counseling during Phase I obviated the need for diet journaling [44, 50] and the burden associated with dietary recall, an effect that later confirmed the efficacy of the dietary education when participants implemented the diet on their own during Phase II. Collectively, the preliminary results of this trial suggest that nutritional ketosis can be attained within the first week on a WFKD and can be sustained in women with MBC who receive proper ketogenic nutrition coaching and guidance.

Thirteen participants (80%) experienced a decrease in fasting glucose after the highly-supervised Phase I. WFKDs are well documented to reduce blood glucose in both healthy and clinical populations, an effect that is a magnitude order greater in patients with hyperglycemia [51–53]. In our trial we observed statistical differences in glucose variability between-phases, which prompted further investigation into predicting long-term dietary success. Almost half of the women who decided to discontinue the study after Phase I (47%) exhibited more than 2-fold glucose variability over three months (± 10% deviation from the mean) compared to participants who completed Phase II. This effect may be partly explained by the self-reported stress and cancer/chemotherapy treatment concurrent with participation in the study, which was largely heterogeneous at baseline.

The chemotherapy drugs used during the study were documented to induce mild hyperglycemic effects, specifically paclitaxel in combination with dexamethasone [48, 54]. We speculate that self-reported stress and drug heterogeneity may collectively be responsible for skewing higher fasting glucose values between women who completed Phase I only and Phase I+II (Δ phase difference: 6%), thereby possibly obfuscating the true glucose effects of WFKD. Although hyperglycemia is a common side-effect in cancer interventions [55], we were able to demonstrate that fasting glucose can significantly decrease after three months of highly-supervised WFKD, concurrent with drug use, and likely mediated by ongoing dietary carbohydrate restriction and weight loss. Due to the nature of our single-arm diet assignment that was informed by prior similar studies [26, 31], the results presented herein must be viewed with the understanding of drug heterogeneity and attrition. Future studies are encouraged to randomize and stratify by drug use, when possible, to provide more conclusive interpretations regarding variability in metabolic outcomes.

Fasting insulin and insulin sensitivity improved in 13 patients (87%) after 3 months, revealing a highly positive outcome for adjuvant cancer treatment. Impaired insulin clearance and insulin resistance are germane to anti-tumor therapy because pharmacological inhibition of insulin-signaling (i.e., GLUT4 stimulation) decreases tumor glucose uptake, thereby limiting substrate availability for growth [14, 54, 56]. While anti-tumor medication helps to restrict tumor proliferation, it can simultaneously downregulate insulin-mediated glucose disposal to skeletal muscle and promote hyperglycemia/hyperinsulinemia [46, 47, 54]. We demonstrated that a WFKD reduced the glycolytic demand of diet as evidenced by the Phase I and II insulin trends, effects that are consistent with previous interventions that employed supervised ketogenic diets [26, 27, 33, 45, 46, 52]. Collectively, ketogenic diets' roles in anti-tumor therapy remain generally unclear [55], however, our intervention demonstrated that a supervised *ad libitum* WFKD consumed for three months can significantly improve insulin sensitivity and mitigate common side-effects induced by anti-tumor medication.

As expected, the *ad libitum* WFKD was associated with significant weight-loss, primarily derived from adipose tissue. Body composition change after Phase I were more favorable than results typically reported in ketogenic diet literature (i.e., significantly greater fat mass to lean mass loss) [57], although in line with body composition changes reported in intervention-based breast cancer studies [58]. The outcomes we presented are positive for several reasons. First, advanced imaging revealed no sign of sarcopenic obesity (loss of lean body mass and gain of fat mass), averting a main pitfall of chemotherapy and dysregulated eating patterns related to cancer treatment [59]. Secondly, reduction of total body fat mass was a desirable outcome with respect to the cancer risks previously described in the Women's Health Initiative study [60] and other ketogenic interventions that used DXA to quantify body composition in breast cancer patients [24]. The pathophysiology of excess adiposity and cancer is increasingly complex; however, reduction of body fat mass has been positively associated with reduction in inflammatory cytokines and bioavailable estradiol, thereby inhibiting tumor growth and

promoting the likelihood of cancer remission [59]. To increase precision in body composition assessments future work should concentrate on using advanced imaging software, such as DXA or MRI, and minimize standardized metrics (i.e., body mass index) to guide dietary success.

Although we intended to conduct a two-arm trial, a limitation of this study is the lack of control or standard of care group, which would have provided context for the normal progression of outcomes in women consuming a non-ketogenic eating pattern recruited from this single study site. During recruitment, there was a unanimous preference to self-select into the WFKD arm and explicit disinterest in the proposed control arm (i.e., low-fat treatment), thus, future randomized trials should be aware of the challenge associated with performing randomized trials within this population. Another limitation is the heterogeneity of chemotherapy treatments and tumor subtypes present in the cohort. Given this was an exploratory study focused on feasibility and efficacy, we wanted to keep it contained to a single site. To meet our enrollment goals, we were not overly restrictive in exclusion criteria by focusing on a specific tumor subtype or chemotherapy drug. Moreover, our feasibility and adherence findings must be viewed with the understanding of our study methods and research sample, and merit attention if extended to the general population.

## Conclusion

Women with MBC undergoing chemotherapy were motivated to participate in a ketogenic intervention trial. Most of them transitioned successfully to a WFKD and demonstrated improved metabolic health consistent with responses observed in non-cancer populations. The WFKD was well tolerated by women who demonstrated high adherence as confirmed by daily capillary blood ketone monitoring and had no adverse diet-related events. After three months, women with MBC experienced significant improvements in fasting blood glucose, insulin, insulin resistance, body weight, and body composition, effects that have been observed in healthy individuals. These beneficial metabolic outcomes were sustained in a subset of women who completed six months of the WFKD. Given these positive outcomes, future research studies should explore the use of a similarly designed WFKD as an adjunct therapy in women with MBC, with emphasis on examining a larger cohort, including a control group, and possibly identifying key characteristics in responders and non-responders. Future studies should also consider including a follow-up protocol that assesses long-term benefits to metabolic health, quality of life, and progression-free survival.

## Supporting information

**S1 Table. Drug prescription.**
(DOCX)

**S2 Table. Menu items.**
(DOCX)

**S1 File. Cancer clinical trial research protocol.**
(DOCX)

**S2 File. TREND checklist.**
(DOCX)

## Acknowledgments

The authors would like to thank the participants, oncologists, research volunteers, and metabolic kitchen staff for their invaluable time and efforts that guaranteed this project's success.

## Author Contributions

**Conceptualization:** Parker N. Hyde, Gerald Krystal, Maryam B. Lustberg, Jeff S. Volek.

**Data curation:** Parker N. Hyde, Jeff S. Volek.

**Formal analysis:** Alex Buga, Teryn N. Sapper, Parker N. Hyde, Brandon Fell, Ryan Dickerson, Katherine Binzel.

**Funding acquisition:** Parker N. Hyde, Jeff S. Volek.

**Investigation:** Parker N. Hyde, Jeff S. Volek.

**Methodology:** Parker N. Hyde, Jeff S. Volek.

**Project administration:** David G. Harper, Jeff S. Volek.

**Supervision:** David G. Harper, Parker N. Hyde, Jeff S. Volek.

**Visualization:** Alex Buga.

**Writing – original draft:** Alex Buga, David G. Harper, Teryn N. Sapper, Jeff S. Volek.

**Writing – review & editing:** Alex Buga, David G. Harper, Teryn N. Sapper, Parker N. Hyde, Justen T. Stoner, Madison L. Kackley, Christopher D. Crabtree, Drew D. Decker, Bradley T. Robinson, Jeff S. Volek.

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
