## [Decision Letter · Decision Letter 0]

16 Oct 2023

PONE-D-23-27894Feasibility and Metabolic Outcomes of a Well-Formulated Ketogenic Diet as an Adjuvant Therapeutic Intervention for Women with Stage IV Metastatic Breast Cancer: The Keto-CARE TrialPLOS ONE

Dear Dr. Volek,

Thank you for submitting your manuscript to PLOS ONE. After careful consideration, we feel that it has merit but does not fully meet PLOS ONE’s publication criteria as it currently stands. Therefore, we invite you to submit a revised version of the manuscript that addresses the points raised during the review process. The manuscript has been evaluated by four reviewers, and their comments are available below.

The reviewers have raised a number of concerns that need attention. They request additional information on methodological aspects of the study (such as the inclusion of information on the sample size and response rate), revisions to the statistical analyses and they question the internal and external validity of the results reported.

Could you please revise the manuscript to carefully address the concerns raised?

We look forward to receiving your revised manuscript.

Kind regards,

Jennifer Tucker, PhD

Staff Editor

PLOS ONE

2. We note that you have selected “Clinical Trial” as your article type. PLOS ONE requires that all clinical trials are registered in an appropriate registry (the WHO list of approved registries is at https://www.who.int/clinical-trials-registry-platform/network/primary-registries"" https://www.who.int/clinical-trials-registry-platform/network/primary-registries" https://www.who.int/clinical-trials-registry-platform/network/primary-registries and more information on trial registration is at http://www.icmje.org/about-icmje/faqs/clinical-trials-registration/). Please state the name of the registry and the registration number (e.g. ISRCTN or ClinicalTrials.gov) in the submission data and on the title page of your manuscript. a) Please provide the complete date range for participant recruitment and follow-up in the methods section of your manuscript. b) If you have not yet registered your trial in an appropriate registry, we now require you to do so and will need confirmation of the trial registry number before we can pass your paper to the next stage of review. Please include in the Methods section of your paper your reasons for not registering this study before enrolment of participants started. Please confirm that all related trials are registered by stating: “The authors confirm that all ongoing and related trials for this drug/intervention are registered”. Please see http://journals.plos.org/plosone/s/submission-guidelines#loc-clinical-trials for our policies on clinical trials.

[I have read the journal's policy and the authors of this manuscript have the following competing interests:

JSV receives royalties from book sales; is a founder and has equity in Virta Health; and is a science advisor for Simply Good Foods and Cook Keto.  MBL has received consulting fees from AstraZeneca, Biotheranostics, Novartis, Pfizer and PledPharma. The remaining authors have no relevant financial or non-financial interests to disclose.]. 

6. We note that Figure 1 and  3 in your submission contain copyrighted images. All PLOS content is published under the Creative Commons Attribution License (CC BY 4.0), which means that the manuscript, images, and Supporting Information files will be freely available online, and any third party is permitted to access, download, copy, distribute, and use these materials in any way, even commercially, with proper attribution. For more information, see our copyright guidelines: http://journals.plos.org/plosone/s/licenses-and-copyright.

A. You may seek permission from the original copyright holder of Figure 1 and  3 to publish the content specifically under the CC BY 4.0 license. 

B. If you are unable to obtain permission from the original copyright holder to publish these figures under the CC BY 4.0 license or if the copyright holder’s requirements are incompatible with the CC BY 4.0 license, please either i) remove the figure or ii) supply a replacement figure that complies with the CC BY 4.0 license. Please check copyright information on all replacement figures and update the figure caption with source information. If applicable, please specify in the figure caption text when a figure is similar but not identical to the original image and is therefore for illustrative purposes only.

Reviewers' comments:

Reviewer's Responses to Questions

**Comments to the Author**

1. Is the manuscript technically sound, and do the data support the conclusions?

Reviewer #1: Yes

Reviewer #2: Yes

Reviewer #3: Yes

Reviewer #4: Yes

2. Has the statistical analysis been performed appropriately and rigorously? 

Reviewer #1: Yes

Reviewer #2: No

Reviewer #3: No

Reviewer #4: I Don't Know

3. Have the authors made all data underlying the findings in their manuscript fully available?

Reviewer #1: Yes

Reviewer #2: Yes

Reviewer #3: No

Reviewer #4: Yes

4. Is the manuscript presented in an intelligible fashion and written in standard English?

Reviewer #1: Yes

Reviewer #2: Yes

Reviewer #3: Yes

Reviewer #4: Yes

5. Review Comments to the Author

Reviewer #1: This manuscript helps to build on a growing body of literature supporting the use of KD as an adjuvant therapy for cancer. As stated, a main strength of this project is the longevity of the intervention, as there is relatively little data beyond 3 months.

-Eligibility criteria: Were any efforts made to limit additional dietary practices (e.g., intermittent fasting or supplement use)?

-Adherence: It seems that the primary means of assessing adherence was daily capillary ketone measurements. Was diet composition assessed during either phase? Were 24-h recalls conducted during phone calls with dietitians? It is clear from the text as is that tracking of nutrients by participants was not required or encouraged, but I am curious to know if diet composition was analyzed by the study team.

-Protein goals: Why was the range 0.6-1.0g/kg selected for protein target?

Reviewer #2: Given the growing popularity of ketogenic diets among cancer patients, and breast cancer patients in particular, this is an important study, because it provides evidence that such a diet can safely and effectively be implemented even in advanced breast cancer patients. The results and discussion are well written and comprehensive. However, I think there are also some points in the current manuscript which can be improved or clarified.

1. Introduction, pages 3/4: Reference 14 is not enough as a reference for all the anti-tumor effects of a KD that you mention. Please cite 1-3 review articles which discuss these effects in more detail.

2. The outcomes of interest are not explicitly mentioned in the Materials and Methods section.

3. Page 8, line 105: Meals were free of cost, but I did not find a statement who sponsored these meals and therefore the study. Please mention the funding sources and their role in the study design.

4. Page 9, line 128: How was a protein intake of 0.6-1 g/kg lean body mass justified?

5. Page 11, line 176: At which time of the day were glucose and ketone measured? Was this specified for the participants?

6. Page 12, Statistics: The name of the test is Shapiro-Wilk, not Wilks. Did you consider an alternative test to the t-test in case of non-normality? What were the “exploratory objectives” you wanted to address? You also mention that you did repeated-measures mixed effects ANOVA followed by post-hoc correction. It would be good to write down an equation showing how this looks like in order to help the reader comprehend which variable is predicted and which are the predictors. Also, I found not results (p-values) of this analysis in the results section.

7. Were there any women with type 2 diabetes included?

Minor comments:

Page 14, line 243: Should be Figure 5, not 4.

Page 16, line 266: Should be Figure 6, not 5.

Page 17, line 288: Should be Figure 7, not 6.

Reviewer #3: 1. Please specify how sample size was estimated.

2. Please identify the secondary outcome.

3. How did you minimize the confounding effects of diet (adjust for carbohydrate or protein and energy intake) or BMI

4. Line 58 and 59 not true

Reviewer #4: This is an important addition to the body of work supporting the use of WFKD as adjuncts to treatment in this and other cancers.

Is it proper form to hyphenate "x-month" when it is not used as a compound adjective?

https://emareye.com/resource-center/hyphens/#:~:text=Hyphens%20are%20often%20used%20to,two%2Dyear%2Dold%20child.

32. Clarify: add 'dietary' = role of dietary fat

38. Is this true even in women with lower BMI than this cohort? (theoretical "lean mass hyper-responder" may demonstrate induced rather than congenital mixed hyperlipidemia)

39. Suggest 'impact' in place of 'effects'

47. Suggest adding 'cancer-related upregulation of phosphoinositol-3 kinase (PI3K) and mammalian target of rapamycin (mTOR) signaling'

93. Hypenate 'two-arm'

98. Clarify: were the women you tried to recruit as controls also offered 3 months of free personalized perpared meals and daily dietetic coaching? If not, this may have been the primary reaon that they chose the WFKD option.

146. I think it's important to mention that there was no need to track intake given that their meals were deemed nutritionally complete by the dietetic team. And without food records, how did you determine if the meals were nutritionally adequate during Phase II?

172. It is unclear what 'pane zucchini cakes' are. Can you simply state 'zucchini cakes?'

233. 'Passed away' (a euphemism) doesn't belong in a research paper. Replace this with 'died' (a term more in line with medical definitions). This needs to be corrected with any mention of 'passed away' in the manuscript and supporting material.

237. Consider restating: 'preference for non-keto desserts as comfort food'

243. Was “fasting” considered to be any time up until their first meal? Or was it suggested that they test within a certain time frame after awakening? Any chance that women could have skipped a day if they believed their ketones might have been impacted by food choices from the prior day? Were their self-reports of ketones/glucose verified by the team, e.g., screen shots of meter readings? Or... by a review of the meter history by a team member at clinic check-ins? My apologies if somehow I missed seeing the details.

249. Consider rewording: 'One participant with hyperglycemia demonstrated a 62 mg/dL decrease in glucose (-34%); the remaining participants demonstrated a more modest but still significant decrease in glucose ()'

267. Body composition improved against what standard? In other words, is a decease in body fat percentage always considered to be an improvement, even in the women with lower BMI at baseline? Or... were you pooling the results rather than suggesting individual improvement? If so, please make that clearer. And perhaps add a mention that in future research, if women with a lower BMI (or more rigorous assessment of body fat percentage) were included, undesirable loss of body fat would need to be more closely monitored by the dietetic team.

269. Suggest rewording: To get around the awkwardness of including a percentage as the first 'word,' perhaps: 'The overall ratio of fat mass to lean body mass lost was 3.5:1 kg:kg, with 78% of the weight loss derived from body fat mass () and 22% from lean body mass ().'

331. Yes! Noteworthy indeed! Consider emphasizing that point here; for example, by adding an adjective, such as 'especially noteworthy' or changing this to 'of utmost importance.' The 2DIRECT trial of women with Her2 negative breast cancer presented findings in a paper comparing FMD and to standard diet: the women in the intervention were not given steroid medication. COnsidering that steroids are routinely administered as part of the SOC, perhaps add a mention of that paper? https://www.nature.com/articles/s41467-020-16138-3

333. I know you understand that you can't manipulate food intake outside of force feeding. Consider changing this to: "Daily capillary BHB objectively assessed dietary adherence and that data was then used by the team to adjust meal composition ()' That also helps to highlight the importance of a diet that was responsive to changing needs of the individuals. This same approach can serve to limit unintended/undesirable weight loss, at times leading to the team's decision to modify the diet even when it means a loss of ketosis. I think this is an extremely important point, often lost in the push of some researchers that advocate any means necessary to maintain a low GKI.

337. Careful here: it sounds like you're saying that the monitoring model was 'obviating the need for diet journaling' when in fact it was your team's constant adjustments to the plan while still providing nutritonally complete meals that obviated the need for journaling.

368. A decrease in serum glucose would also result in a dampened insulin response, reducing non-pharm insulin signaling. This is important given the increase in the number of insulin receptors found on cancer cells.

370. You reference a paper on the impact of capecitabine. A strongly recommend that you add a reference to at least one paper demonstrating that same effect on women receiving PI3Ki.

392. Excellent! Changing to another assessment for LBM might also lead to less reluctance to include women with low BMI that are lean and metabolically healthy (other than their cancer). You may have noticed that this is more often true in women with TNBC. And in fact, it is seen here in your study with 2 of the 3 women with TNBC at the lowest BMI.

397. In future research, consider how to use language to present the control arm in a more positive light. For example, refering to it as a Mediterranean diet might be a draw considering the current emphasis on the perceived health benefits surrounding that term.

407. 'Motivated' is key here to the initial choice of a diet. Did you explore why they opted for the WFKD? If so, their input could help in the design of future studies. As you are well aware, failure to recruit adequate numbers of participants has doomed many other trials right at the start.

414. As you know, retention is extremely important for future clinical trials. Did you attempt to identify (or care to speculate on) the characteristics in the subset of women who completed 6 months?

417. Somewhat awkward ending here. How about, '... examining a larger cohort that includes a control group and attempts to identifiy key characteristics of both responders and non-responders. Future studies should also consider including a follow-up protocol that assesses long-term benefits to metabolic health, quality of life, and progression-free survival.

Questions:

Was MCT oil suggested? If not, why not?

Did the number of meals/snacks vary according to patient preferences?

In your Main Takeaways sidebar, you state that 'women improved metabolic health as evidenced by decreased body fat.' I think this is somewhat problematic for reasons I've stated above.

Although you don't menition supplementation, we know from real-world experience that many people in cancer treatment do include anywhere from a few to way too many. Care to include some mention on whether or not you believe this could have some impact on ketosis (e.g., ketone salts)? or. lowering glucose (e.g., berberine)?

Although this is outside of the scope of your study, did you compare CBC/CMP values to historical controls? It would be interesting to learn what if any differences you saw, eg Na+ levels, platelets, inflammatory markers.

Did I miss a list of the medications that women were taking at baseline or throughout the study? For example, statin drugs are known to raise glucose (and suppress ketones) in a subset of people. I would also like to see the amount of dex that was included for each woman. That might factor into future studies of subtypes where dex is mandatory.

Feasibility in the general population cannot be assumed from this study. Have you adequately addressed that here?

Did you consider any of these references?

https://www.frontiersin.org/articles/10.3389/fnins.2020.00390/full

https://link.springer.com/article/10.1007/s11060-020-03417-8

https://ascopubs.org/doi/abs/10.1200/JCO.2016.34.15_suppl.e23173

6. PLOS authors have the option to publish the peer review history of their article (what does this mean?). If published, this will include your full peer review and any attached files.

Reviewer #1: No

Reviewer #2: **Yes: **Rainer J. Klement

Reviewer #3: No

Reviewer #4: No

---

## [Author Response · Author response to Decision Letter 0]

24 Oct 2023

Please refer to the "Response to Reviewers_PONE-D-23-27894" document for detail regarding reviewer/editor responses.

---

## [Editor Report · Decision Letter 1]

21 Nov 2023

PONE-D-23-27894R1Feasibility and Metabolic Outcomes of a Well-Formulated Ketogenic Diet as an Adjuvant Therapeutic Intervention for Women with Stage IV Metastatic Breast Cancer: The Keto-CARE TrialPLOS ONE

Dear Dr. Volek,

Thank you for submitting your revised manuscript to PLOS ONE. After careful consideration, we feel that it does not yet fully meet PLOS ONE’s publication criteria as it currently stands. Therefore, we invite you to submit a revised version of the manuscript that addresses the points raised during the review process.

We look forward to receiving your revised manuscript.

Kind regards,

Rainer J. Klement, Ph.D.

Guest Editor

PLOS ONE

Journal Requirements:

Additional Editor Comments:

Thank you very carefully addressing all comments provided by the reviewers. The quality of the manuscript has improved sufficiently to get published in principal, but there are two points remaining: In your response to reviewer #2 who advised to provide more statistical details you mentioned referred to the publicly available dataset. However, the link you provided is not working or does not exist, respectively (DOI 10.5061/dryad.kh18932d4). Furthermore, please specify more precisesly the sentence on line 226-229 "...we analyzed the main effects of time using dependent samples t-tests..." -> the main effects of time on which variables?

---

## [Author Response · Author response to Decision Letter 1]

4 Dec 2023

Please refer to "Response to Reviewers_PONE-D-23-27894R1" for additional details.

---

## [Decision Letter · Decision Letter 2]

11 Dec 2023

PONE-D-23-27894R2Feasibility and Metabolic Outcomes of a Well-Formulated Ketogenic Diet as an Adjuvant Therapeutic Intervention for Women with Stage IV Metastatic Breast Cancer: The Keto-CARE Trial

PLOS ONE

Dear Dr. Volek,

Thank you again for making additional revisons to your manuscript according to the reviewer comments. In the meanwhile, however, a fifth reviewer with expertise in biostatistics has been assigned by the Staff Editor to review the manuscript. The comments of this reviewer are attached below and should be considered before we can make a final decision about the publication of your study.

I would be happy if you could adress these additional points and submit a newly revised verison of your manuscript.

We look forward to receiving your revised manuscript.

Kind regards,

Rainer J. Klement, Ph.D.

Guest Editor

PLOS ONE

Journal Requirements:

Reviewers' comments:

Reviewer's Responses to Questions

**Comments to the Author**

1. If the authors have adequately addressed your comments raised in a previous round of review and you feel that this manuscript is now acceptable for publication, you may indicate that here to bypass the “Comments to the Author” section, enter your conflict of interest statement in the “Confidential to Editor” section, and submit your "Accept" recommendation.

Reviewer #5: (No Response)

2. Is the manuscript technically sound, and do the data support the conclusions?

Reviewer #5: Partly

3. Has the statistical analysis been performed appropriately and rigorously? 

Reviewer #5: No

4. Have the authors made all data underlying the findings in their manuscript fully available?

Reviewer #5: Yes

5. Is the manuscript presented in an intelligible fashion and written in standard English?

Reviewer #5: Yes

6. Review Comments to the Author

Reviewer #5: This manuscript reports a single arm clinical trial investigating feasibility and Metabolic outcomes of a well-formulated Ketogenic diet as an adjuvant therapeutic intervention for women with stage IV metastatic breast cancer. This design of this clinical trial is revised from a two arms study to a single arm study due to hard to recruit controls during pandemic. I have below comments.

Please provide statistical considerations about sample size for this study.

For the data shown in the uploaded Prism file, data needs to be confirmed. For example,

Data at 3 MO in “Ketones” are different from data at week 12 in “*Individual Ketone Data”. They should have been consistent.

In “*Clean Individual Ketone Data”, Partial has 8 samples, 5 of them have data at 3MO, which is different from n=6 described in manuscript.

In the manuscript, it reported 9 Completers and 6 Partial. But in data from “*T-test BHB Partial vs. Completers”, both Completers and Partial have same N=13.

It would be helpful to have one biostatistician in the team to reanalyze the data and confirm the results.

7. PLOS authors have the option to publish the peer review history of their article (what does this mean?). If published, this will include your full peer review and any attached files.

Reviewer #5: No

---

## [Author Response · Author response to Decision Letter 2]

12 Dec 2023

Please refer to the attached document: "Response to Reviewers_PONE-D-23-27894R2"

Additional comment: the PRISM Dryad dataset was updated to reflect Reviewer 5 biostatistics comments.

---

## [Editor Report · Decision Letter 3]

14 Dec 2023

Feasibility and Metabolic Outcomes of a Well-Formulated Ketogenic Diet as an Adjuvant Therapeutic Intervention for Women with Stage IV Metastatic Breast Cancer: The Keto-CARE Trial

PONE-D-23-27894R3

Dear Dr. Volek,

We’re pleased to inform you that your manuscript has been judged scientifically suitable for publication and will be formally accepted for publication once it meets all outstanding technical requirements.

Kind regards,

Rainer J. Klement, Ph.D.

Guest Editor

PLOS ONE

Additional Editor Comments (optional):

Thank you for adressing the remaining comments about the statistics. The paper is now suitable for publication.
---

## [Editor Report · Acceptance letter]

21 Dec 2023

PONE-D-23-27894R3 

PLOS ONE

Dear Dr. Volek, 

I'm pleased to inform you that your manuscript has been deemed suitable for publication in PLOS ONE. Congratulations! Your manuscript is now being handed over to our production team.

Kind regards, 

on behalf of

Dr. Rainer J. Klement 

Guest Editor

PLOS ONE